# Post-Transplantation Seroprotection Rates in Liver, Lung, and Heart Transplant Recipients Vaccinated Pre-Transplantation against Hepatitis B Virus and Invasive Pneumococcal Disease

**DOI:** 10.3390/vaccines12101092

**Published:** 2024-09-25

**Authors:** Lise Bank Hornung, Sebastian Rask Hamm, Annemette Hald, Zitta Barrella Harboe, Lene Fogt Lundbo, Neval Ete Wareham, Line Dam Heftdal, Christina Ekenberg, Stephanie Bjerrum, Jon Gitz Holler, Inger Hee Mabuza Mathiesen, Paul Suno Krohn, Peter Nissen Bjerring, Finn Gustafsson, Michael Perch, Allan Rasmussen, Susanne Dam Nielsen

**Affiliations:** 1Viro-Immunology Research Unit, Department of Infectious Diseases, Copenhagen University Hospital—Rigshospitalet, 2100 Copenhagen, Denmarksebastian.rask.hamm.02@regionh.dk (S.R.H.);; 2Department of Pulmonary and Infectious Diseases, Copenhagen University Hospital—North Zealand, 3400 Hillerød, Denmark; 3Department of Clinical Medicine, University of Copenhagen, 2200 Copenhagen, Denmark; 4Department of Infectious Diseases, Copenhagen University Hospital—Rigshospitalet, 2100 Copenhagen, Denmark; 5Department of Surgical Gastroenterology and Transplantation, Copenhagen University Hospital—Rigshospitalet, 2100 Copenhagen, Denmark; 6Department of Intestinal Failure and Liver Diseases, Copenhagen University Hospital—Rigshospitalet, 2100 Copenhagen, Denmark; 7Department of Cardiology, Heart and Lung Transplant Unit, Copenhagen University Hospital—Rigshospitalet, 2100 Copenhagen, Denmark

**Keywords:** solid organ transplantation, hepatitis B, invasive pneumococcal disease, vaccination

## Abstract

Vaccination before solid organ transplantation is recommended since post-transplantation immunosuppression is known to impair vaccine responses. However, little is known about post-transplantation seroprotection rates in organ transplant recipients vaccinated pre-transplantation. We aimed to investigate the proportion of transplant recipients vaccinated against hepatitis B virus (HBV) and invasive pneumococcal disease (IPD) pre-transplantation at the time of listing for transplantation with post-transplantation seroprotection. We included 136 solid organ transplant (SOT) recipients vaccinated at the time of listing for transplantation. We investigated post-transplantation antibody concentrations against HBV and IPD. Established antibody thresholds were used to define seroprotection. The proportions of SOT recipients with post-transplantation seroprotection were 27.9% (*n* = 38) and 42.6% (*n* = 58) against HBV and IPD, respectively. Compared to completing HBV vaccination pre-transplantation, completing post-transplantation vaccination (adjusted odds ratio (aOR): 7.8, 95% CI: 2.5–24.5, *p* < 0.001) and incomplete vaccination (aOR: 6.3, 95% CI: 1.2–32.6, *p* = 0.028) were associated with non-response against HBV, after adjustment for confounders. Importantly, patients with seroprotection at the time of listing had lower odds of non-response against HBV (aOR: 0.04, 95% CI: 0.0–0.1, *p* < 0.001) and IPD (aOR: 0.3, 95% CI: 0.1–0.7, *p* = 0.007) compared to those without seroprotection. SOT recipients vaccinated pre-transplantation had low post-transplantation seroprotection rates against HBV and IPD. However, SOT recipients with seroprotection at the time of listing had lower odds of non-response, suggesting early vaccination should be a priority.

## 1. Introduction

Organ transplant recipients receive immunosuppressive therapy to prevent allograft rejection after transplantation, which, in turn, increases susceptibility to and risk of complications from infections [1]. Specifically, the incidence of invasive pneumococcal disease (IPD) is more than seven times higher among solid organ transplant (SOT) recipients than in the general population [2]. While de novo hepatitis B virus (HBV) infection in SOT-recipients is rare, it has been shown to reduce graft survival in liver transplant recipients [3,4]. Both HBV and IPD are vaccine-preventable diseases [5,6]. Unique for these diseases are known antibody concentration thresholds for protection against disease. In transplant recipients, reduced antibody responses are seen after pneumococcal vaccination, and HBV vaccine-response rates are lower in liver transplant recipients vaccinated after transplantation than in candidates vaccinated before [7,8]. It is therefore recommended in several guidelines to vaccinate transplant candidates prior to transplantation [9,10,11]. However, there are limited data to support this strategy [12,13]. In contrast, previous studies have shown response rates between 13.0% and 67.0% to HBV vaccination in patients with chronic liver disease who are on the transplant wait list [14,15]. While some studies suggest higher response rates among transplant candidates when vaccinated against HBV during early stages of liver disease [16], there are limited data on post-transplantation seroprotection rates when transplant candidates are vaccinated against HBV and IPD pre-transplantation [17,18,19,20]. Furthermore, previous studies were either conducted more than two decades ago, evaluated heterogeneous vaccination schedules, or included few participants [18,19,20]. Thus, there are important gaps in the knowledge of the effect of pre-transplantation vaccination.

This study was conducted to provide information to guide vaccination strategies. We aimed to determine the proportion of transplant recipients vaccinated against HBV and IPD pre-transplantation at the time of listing for transplantation with post-transplantation seroprotection. Furthermore, we determined risk factors for responses below the protective threshold (non-response).

## 2. Methods

### 2.1. Study Design and Population

We conducted a prospective cohort study including adult SOT recipients (≥18 years) vaccinated at the time of listing for transplantation between February 2020 and October 2022 before undergoing liver, lung, or heart transplantation. In Denmark, all liver and lung transplantations and around half of heart transplantations are performed at Copenhagen University Hospital, Rigshospitalet, where transplant candidates are assessed at a centralized pre-transplantation vaccination clinic at Department of Infectious Diseases. This is a structured vaccination program based on serology, exposures, and vaccination history for HBV and IPD, with >95% adherence of transplant candidates at our center [21].

### 2.2. Pre-Transplantation Serology and Vaccination

Before assessment at the vaccination clinic, the SOT candidates underwent serological screening for HBV surface antibodies (anti-HBs) and pneumococcal antibodies.

Candidates with anti-HBs concentrations above the protective cut-off and/or documented history of complete HBV vaccination were offered a booster vaccine. A full HBV vaccination schedule of three doses of 20 µg recombinant HBV surface antigen administered at 0, 1, and 6 months was offered to candidates without seroprotection or documentation for completed HBV vaccination. The available vaccines were Twinrix^®^ (GlaxoSmithKline, London, UK), Engerix-B^®^ (GlaxoSmithKline, London, UK) and Fendrix^®^ (GlaxoSmithKline, London, UK).

Candidates were vaccinated against IPD based on their vaccination history and pneumococcal serology. The recommended pneumococcal vaccinations were either one 13-valent pneumococcal conjugate vaccine (PCV13) followed by a 23-valent pneumococcal polysaccharide vaccine (PPSV23) after 8 weeks or one dose of 20-valent pneumococcal conjugate vaccine (PCV20). If the patient had previously received PPSV23, this was followed by one dose of PCV13 or PCV20 at least one year after the PPSV23 dose. The available vaccines were Pneumovax 23^®^ (Merck Sharp & Dohme Corp. Rahway, NJ, USA), Prevenar 13^®^ (Pfizer Europe MA EEIG, Brussels, Belgium), and Apexxnar^®^ (Pfizer Europe MA EEIG, Brussels, Belgium).

### 2.3. Post-Transplantation Serology

Concentrations of anti-HBs and the geometric mean concentration (GMC) of antibodies against 12 representative pneumococcal serotypes were measured after transplantation as a part of routine clinical care. Anti-HBs were analyzed following local instructions, and pneumococcal antibodies were quantified at Statens Serum Institute (SSI) using in-house Luminex technology. If more than one test result was available, we used the earliest serology measured at least three months after transplantation.

### 2.4. Data Retrieval

Information about vaccination history, including timing of vaccines, was retrieved from the Danish Vaccination Registry (DDV) [22]. DDV has complete national coverage and contains all vaccines administered in Denmark since 2015.

Antibody test results were collected from the Danish Microbiology Database (MiBa), a nationwide database containing all microbiology results in Denmark since 2011 [23], and from medical records. Clinical data including immunosuppressive therapy and acute rejections were extracted from medical records.

The study was conducted following the Declaration of Helsinki. According to Danish legislation, the retrieval of data without collection of informed consent was approved by the institutional review board at the Center for Regional Development (R-20051155).

### 2.5. Definitions

The WHO has defined an anti-HBs concentration of >10 IU/L as protective against HBV [24]. A pneumococcal antibody concentration of ≥0.35 µg/mL is considered protective against pneumococcal disease from a specific serotype in children [24]. However, since thresholds vary by pneumococcal serotype, and a more conservative cut-off has been suggested for adults, we defined seroprotection against IPD as a GMC of IgG antibodies against 12 representative serotypes of ≥1 µg/mL [20,25].

Completed HBV vaccination pre-transplantation was defined as three doses administered pre-transplantation. Post-transplantation completion was defined as ≥1 of the three doses administered after transplantation. Individuals with seroprotection at the time of listing for transplantation who received one HBV booster vaccination were defined as protected at the time of listing and boosted.

Completed IPD vaccination pre-transplantation was defined as either having seroprotection at the time of listing or having received PCV13 and PPSV23 or one dose of PCV20 administered pre-transplantation. Completed post-transplantation IPD vaccination was defined as ≥1 vaccine administered after transplantation.

The time of completion of a vaccine series was defined as the date of the last vaccine.

### 2.6. Statistical Analysis

Continuous data were reported as means or medians and compared using Student’s *t*-test or Mann–Whitney U test, as appropriate. Categorical data were reported as percentages and compared using Pearson’s chi-square test or Fisher’s test, as appropriate.

Seroprotection rates were reported as proportions. Logistic regression models with vaccine seroprotection against either HBV or IPD as the dependent variable were used to explore risk factors for vaccine non-response. In adjusted models, variables were tested one at a time, adjusting for age, sex, and transplant type. Furthermore, a sensitivity analysis of the logistic regression models, including serology status at the time of listing for transplantation, was performed for the adjusted models.

## 3. Results

### 3.1. Cohort Characteristics

We included 136 SOT recipients with pre-transplantation vaccination and a post-transplantation HBV and IPD serology test result (Figure 1). Of those, 87 (64.0%) were liver, 34 (25.0%) were lung, and 15 (11.0%) were heart transplant recipients (clinical characteristics shown in Table 1).

HBV serology at the time of listing for transplantation was available for 122 (89.7%) SOT recipients, and of those, 23 (18.9%) had existing seroprotection against HBV (Figure 2). IPD serology from the time of listing for transplantation was available for 132 (97.1%) SOT recipients, and of those, 43 (32.6%) had existing seroprotection against IPD (Figure 2).

### 3.2. Hepatitis B Virus

Post-transplantation protective anti-HBs concentrations were present in 38 (27.9%) of 136 SOT recipients (Figure 2). Among individuals with post-transplantation seroprotection, 18 (56.2%) had seroprotection at the time of listing for transplantation, while among those without post-transplantation seroprotection, 5 (5.6%, *p* < 0.001) had seroprotection at the time of listing for transplantation. A higher proportion of the SOT recipients with post-transplantation seroprotection had completed their vaccination schedule pre-transplantation compared to those without post-transplantation seroprotection: 19 (50.0%) vs. 22 (22.4%, *p* < 0.001) (clinical characteristics between SOT recipients with and without seroprotection against HBV shown in Table 2).

In a univariable logistic regression model, the odds ratio (OR) for non-response was 9.2 ([95% CI: 3.0–27.6], *p* < 0.001) when completing the HBV vaccination schedule after transplantation and 6.9 ([95% CI: 1.4–34.0], *p* = 0.017) when never completing vaccination compared to SOT recipients who completed vaccination pre-transplantation. Furthermore, SOT recipients with a longer time from assessment at the vaccination clinic to transplantation had an OR of 0.7 ([95% CI: 0.6–0.9], *p* = 0.008) for non-response per 3 months of increase in time from assessment at the vaccination clinic to transplantation.

In a multivariable model, adjusting for age, sex, and type of transplanted organ, the adjusted OR (aOR) for non-response was 7.8 ([95% CI: 2.5–24.5], *p* < 0.001) when completing HBV vaccination schedule after transplantation and 6.3 ([95% CI: 1.2–32.6], *p* = 0.028) when never completing vaccination compared to SOT recipients who completed vaccination pre-transplantation. SOT recipients with a longer time from assessment at the vaccination clinic to transplantation had an aOR of 0.8 ([95% CI: 0.6–1.0], *p* = 0.043) for non-response per 3 months of increase in time from assessment to transplantation (Table 3).

A sensitivity analysis was performed including only the 122 participants with HBV serology at time of listing. In a univariable logistic regression model, SOT recipients with seroprotection against HBV at time of listing had an OR of 0.1 ([95% CI: 0.0–0.1], *p* < 0.001) for non-response compared to those without seroprotection at time of listing.

In a multivariable model adjusted for age, sex, and transplant type, SOT recipients with seroprotection against HBV at time of listing had an aOR of 0.04 ([95% CI: 0.0–0.1], *p* < 0.001) for post-transplantation HBV non-response compared to those without seroprotection at time of listing (Table 4).

### 3.3. Invasive Pneumococcal Disease

Protective antibody concentrations against IPD after transplantation were present in 58 (42.6%) of 136 SOT recipients (Figure 2). Among individuals with post-transplantation seroprotection, 25 (43.9%) had seroprotection at the time of listing for transplantation, while among those without post-transplantation seroprotection, 18 (24.0%, *p* = 0.026) had seroprotection at the time of listing for transplantation. A larger proportion of the SOT recipients with post-transplantation seroprotection had completed their vaccination schedule pre-transplantation compared to those without post-transplantation seroprotection: 44 (75.9%) vs. 52 (66.7%, *p* = 0.008) (clinical characteristics between SOT recipients with and without seroprotection against IPD shown in Table 5).

In a univariable logistic regression model, male SOT recipients had an odds ratio (OR) for non-response of 0.5 ([95% CI: 0.2–0.9], *p* = 0.028) compared to female SOT recipients. No statistically significant risk factors for non-response after vaccination against IPD were found in the multivariable logistic regression model (Table 6).

A sensitivity analysis was performed including only the 132 participants with IPD serology at the time of listing. In a univariable logistic regression model, SOT recipients with seroprotection against IPD at the time of listing had an OR of 0.4 ([95% CI: 0.2–0.9], *p* = 0.017) for post-transplantation IPD non-response compared to those without seroprotection at the time of listing.

In a multivariable model adjusted for age, sex, and type of transplanted organ, SOT recipients with seroprotection against IPD at the time of listing had an aOR of 0.3 ([95% CI: 0.1–0.7], *p* = 0.007) for non-response compared to those without seroprotection at time of listing (Table 4).

## 4. Discussion

In this cohort study, we included 136 solid organ transplant recipients who were referred for pre-transplantation vaccination, including HBV and IPD, at the time of listing for transplantation. Post-transplantation seroprotection rates for HBV and IPD were 27.9% and 42.6%, respectively. Risk factors for HBV non-response were post-transplantation completed HBV vaccination, incomplete HBV vaccination, and shorter time from assessment at the vaccination clinic to transplantation. Importantly, having seroprotection at the time of listing for transplantation was associated with lower odds of non-response for both HBV and IPD.

The post-transplantation HBV seroprotection rate of 27.9% is low, but it is comparable with previous studies of SOT candidates vaccinated against HBV pre-transplantation and demonstrating post-transplantation anti-HBs seroprotection rates from 8.0% to 74.5% [7,17,19]. We found that completion of post-transplantation HBV vaccination or never completing vaccination were associated with higher odds of non-response, and similar findings have been reported before [19]. Thus, Arslan et al. found that liver transplant candidates who completed HBV vaccination pre-transplantation were more likely to elicit detectable HBV antibodies than candidates who completed post-transplantation vaccination [19]. These results support pre-transplantation vaccination of SOT candidates, and underline the importance of not only initiating but completing the schedule prior to transplantation.

Furthermore, we found an association between longer time from assessment at the vaccination clinic to transplantation and lower risk of HBV non-response. This could be due to increased likelihood of completing the vaccination schedule before transplantation. Another explanation could be an association between longer time from assessment at the vaccination clinic to transplantation and less severe organ disease, which has been described to be associated with lower risk of non-response in liver transplant recipients [19].

Interestingly, we found the presence of protective anti-HBs concentrations at the time of listing for transplantation to be strongly associated with lower odds of post-transplantation non-response. This has not previously been described, mainly due to exclusion of patients with seroprotection in many of the previous studies [7,14,15,17,19]. However, one study including kidney transplant patients with pre-transplantation seroprotection against HBV found a post-transplantation seroprotection rate of 86% among those negative for hepatitis B core antibodies [26]. This study and our findings indicate that pre-transplantation seroprotection may be an important factor when aiming to achieve seroprotection after transplantation.

The SOT recipients included in our study had a seroprotection rate of 18.9% against HBV at the time of listing, most likely due to low rates of prior HBV vaccination. Our findings suggest that vaccinating even earlier in the course of the disease might be important to ensure post-transplantation seroprotection, and perhaps, vaccination should take place before the candidate is listed for transplantation. A suggestion for further investigations is double-dose HBV vaccination, which, in a randomized controlled trial, had an OR for seroprotection of 2.48 compared to a single dose when administered to 107 non-responding patients with human immunodeficiency virus (HIV) [27].

The post-transplantation seroprotection rate against IPD was 42.6%, which seems to be low. However, it is difficult to compare with previous studies in SOT recipients due to heterogenous vaccination schedules and different definitions of a positive vaccine response [18,20,28,29]. One trial included kidney transplant candidates and recipients vaccinated against IPD with either a single or double dose of PCV13 and PPSV23 administered with 12 weeks interval. This study found seroprotection rates of 33% among the wait list patients, of whom 51.7% underwent kidney transplantation during the study, and 16% among the transplant recipients 18 months after vaccination [30]. We did not investigate pre-transplantation seroprotection rates after vaccination, but previous studies of vaccination against IPD in liver transplant candidates have reported the waning of antibodies to or below baseline concentrations after transplantation [18,20]. The seroprotection rate against IPD at the time of listing for transplantation was 32.6%, which is higher than for HBV, probably due to IPD vaccination prior to listing for transplantation being more common than HBV vaccination. We found that seroprotection against IPD at the time of listing for transplantation was associated with lower odds of IPD non-response, which supports the recommendation of pre-transplantation vaccination and suggest that vaccination early during the work-up before listing for transplantation may improve the chance of obtaining post-transplantation seroprotection.

The strengths of this study include established protective antibody thresholds for both vaccines and a well-described cohort with detailed data on vaccination history, immunosuppressive therapy, serology, and comorbidities. Our study also has possible limitations. Information on pre-transplantation serology after vaccination was not available, and therefore, we could not investigate the proportion of patients who were seroprotected after vaccination but lost their antibodies after transplantation. While antibodies are important for evaluating seroprotection, our study did not include measuring cellular immunity. Future studies to build on the present study should include pre-transplantation serology after vaccination, more detailed immunology, and, optimally, kidney transplant recipients, who were not included in this study.

## 5. Conclusions

In conclusion, post-transplantation seroprotection rates for both HBV and IPD were low. Risk factors for HBV non-response were completing vaccination after transplantation, having incomplete vaccination, and shorter duration from assessment at the vaccination clinic to transplantation. For both HBV and IPD, we found that patients with seroprotection at the time of listing for transplantation had lower odds of non-response compared to those without seroprotection. Thus, to improve post-transplantation seroprotection rates, we recommend that early vaccination during the pre-transplantation work-up should be a priority. Furthermore, increased attention to follow-up serology and the use of a booster vaccination may be warranted.

## 6. Summary of Key Findings

⇒Post-transplantation seroprotection rates for both hepatitis B virus (HBV) and invasive pneumococcal disease (IPD) were 27.9% (*n* = 38) and 42.6% (*n* = 58) against HBV and IPD, respectively;⇒Risk factors for HBV non-response were completed HBV vaccination post-transplantation, incomplete HBV vaccination, and shorter time from assessment at the vaccination clinic to transplantation;⇒SOT recipients with seroprotection at the time of listing for organ transplantation had lower odds of non-response against HBV and IPD.

## Figures and Tables

**Figure 1 vaccines-12-01092-f001:**
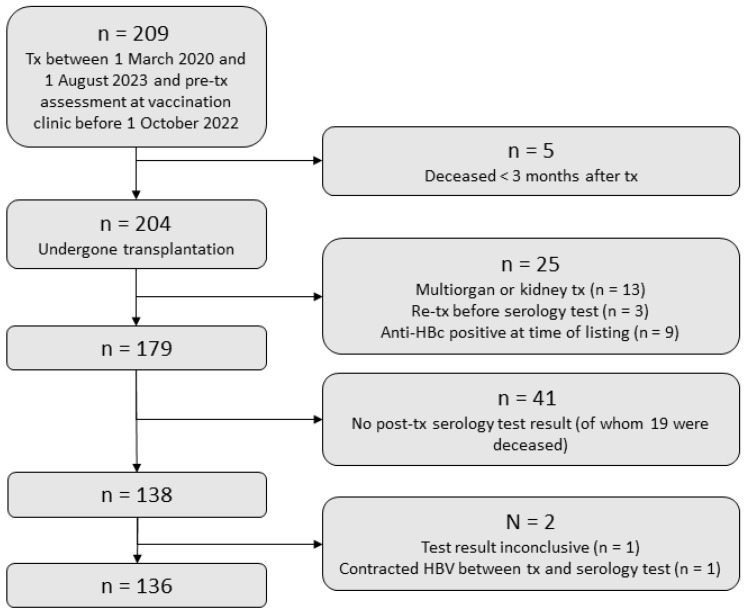
Study flowchart. Tx, transplantation; Anti-HBc, hepatitis B core antibodies; HBV, hepatitis B virus; IPD, invasive pneumococcal disease.

**Figure 2 vaccines-12-01092-f002:**
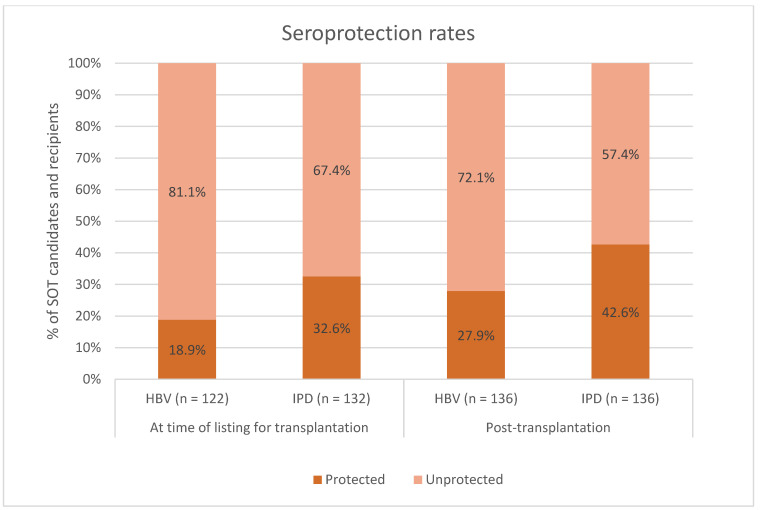
Seroprotection rates among SOT candidates and recipients vaccinated against hepatitis B virus (HBV) and invasive pneumococcal disease (IPD) at time of listing for transplantation. Dark orange: percentage of SOT candidates and recipients with seroprotection. Light orange: percentage of SOT candidates and recipients without seroprotection. *X*-axis: seroprotection rates at the time of listing for transplantation (two columns on the left) and seroprotection rates after transplantation (two columns on the right). *Y*-axis: percentage of SOT candidates and recipients. SOT, solid organ transplant.

**Table 1 vaccines-12-01092-t001:** Patient characteristics at time of listing for transplantation.

	Total (*n* = 136)
Age (median (IQR) in years)	53 (45.8, 59.2)
Male sex, *n* (%)	79 (58.1)
Transplant type, *n* (%)	
Liver	87 (64.0)
Lung	34 (25.0)
Heart	15 (11.0)
Re-transplantation, *n* (%)	7 (5.1)
Anti-HBs, *n* (%)	
Protected	23 (18.9)
Unprotected	99 (81.1)
Missing	14
Pneumococcal antibodies, *n* (%)	
Protected	43 (32.6)
Unprotected	89 (67.4)
Missing	4
Immunosuppressive therapy, *n* (%) (yes/no)	44 (32.4)
Prednisolon	28 (20.6)
Mycophenolate mofetil	12 (8.8)
Azathioprine	11 (8.1)
Tacrolimus	11 (8.1)
Other (cyclosporine, chemotherapy, ibrutinib, TNF-α-inhibitor)	6 (4.4)
Comorbidities, *n* (%)	
Diabetes mellitus	16 (11.8)
Cancer	21 (15.4)
Dialysis	<5

IQR, interquartile range; anti-HBs, hepatitis B surface antibodies.

**Table 2 vaccines-12-01092-t002:** Serology for HBV surface antibodies.

	Protected (*n* = 38)	Unprotected (*n* = 98)	*p*-Value *
Age at serology (median (IQR) in years)	52.5 (35.0, 58.0)	57.5 (50.2, 62.0)	0.015
Age at time of listing for transplantation (median (IQR) in years)	51 (33.2, 56.8)	55 (48.0, 60.0)	0.015
Male sex, *n* (%)	24 (63.2)	55 (56.1)	0.581
Transplant type, *n* (%)			0.491
Liver	22 (57.9)	65 (66.3)	
Lung	10 (26.3)	24 (24.5)	
Heart	6 (15.8)	9 (9.2)	
Re-transplantation, *n* (%)	<5	5 (5.1%)	NA
Anti-HBs at time of listing for transplantation, *n* (%)			<0.001
Unprotected	14 (43.8)	85 (94.4)	
Protected	18 (56.2)	5 (5.6)	
Missing	6	8	
HBV vaccination status, *n* (%)			<0.001
Completed pre-transplantation	19 (50.0)	22 (22.4)	
Completed after transplantation	5 (13.2)	53 (54.1)	
Incomplete	<5	16 (16.3)	
Protected at time of listing and boosted	12 (31.6)	7 (7.1)	
Time from transplantation to serology (median (IQR) in days)	394 (234.5, 605.0)	479.5 (298.2, 668.5)	0.127
Time from vaccination to serology (median (IQR) in days)	610.5 (453.5, 771.8)	607.5 (469.0, 754.5)	0.888
Time from vaccination to transplantation (median (IQR) in days)	160 (91.5, 338.0)	101 (40.0, 194.8)	0.006
Comorbidities after transplantation, *n* (%)			
Diabetes mellitus	<5	24 (24.5)	NA
Dialysis	<5	<5	NA
Cancer	<5	<5	NA
Immunosuppressive therapy			
Immunosupp. at time of listing for transplantation, *n* (%)	13 (34.2)	31 (31.6)	0.933
Induction therapy, *n* (%)			0.926
No induction	22 (57.9)	55 (56.1)	
Simulect/basiliximab	6 (15.8)	22 (22.4)	
ATG	10 (26.3)	21 (21.4)	
Maintenance immunosuppresive therapy, *n* (%)			
Corticosteroid	30 (78.9)	70 (71.4)	0.500
Tacrolimus	28 (73.7)	76 (77.6)	0.801
Cyclosporine	9 (23.7)	16 (16.3)	0.346
mTOR-inhibitors	<5	14 (14.3)	NA
Antimetabolites			0.665
Azathioprine	<5	<5	
Mycophenolate mofetil	33 (86.8)	81 (82.7)	
None	<5	13 (13.3)	
Rejection treated with pulse-steroid, *n* (%)	7 (18.4)	14 (14.3)	0.738

NA, not applicable; HBV, hepatitis B virus; IQR, interquartile range; anti-HBs, hepatitis B surface antibodies; ATG, antithymocyte globulin. * Calculated by chi-square test or Mann–Whitney U test.

**Table 3 vaccines-12-01092-t003:** Risk factors for non-response to HBV vaccination.

	Unadjusted Model			Adjusted Model *		
	OR (Crude)	95% CI	*p*-Value	OR (Adjusted)	95% CI	*p*-Value
Age at time of listing for transplantation per 10-year increase	1.5	1.1–2.1	0.009			
Male sex	0.7	0.4–1.6	0.456			
Transplant type						
Liver	Ref.	Ref.	Ref.			
Heart	0.5	0.2–1.6	0.244			
Lung	0.8	0.3–2.0	0.644			
Time variables						
Time from transplantation to serology per year	1.4	0.8–2.3	0.262	1.4	0.8–2.4	0.260
Time from vaccination to transplantation per 3 months	0.7	0.6–0.9	0.008	0.8	0.6–1.0	0.043
HBV vaccination status						
Completed pre-transplantation	Ref.	Ref.	Ref.	Ref.	Ref.	Ref.
Completed after transplantation	9.2	3.0–27.6	<0.001	7.8	2.5–24.5	<0.001
Incomplete	6.9	1.4–34.0	0.017	6.3	1.2–32.6	0.028
Protected at time of listing and boosted	0.5	0.2–1.5	0.229	0.4	0.1–1.4	0.166
Comorbidities after transplantation						
Diabetes mellitus	3.8	1.1–13.4	0.039	3.5	0.9–13.4	0.068
Immunosuppres. therapy after transplantation						
Antimetabolites						
None	Ref.	Ref.	Ref.	Ref.	Ref.	Ref.
Azathioprine	0.5	0.1–3.8	0.473	0.6	0.1–6.1	0.705
Mycophenolate mofetil	0.6	0.2–2.1	0.398	0.4	0.1–1.9	0.270
Calcineurin inhibitors						
Tacrolimus	1.2	0.5–2.9	0.644	1.2	0.4–3.4	0.786
Cyclosporine	0.6	0.3–1.6	0.323	0.6	0.2–2.0	0.409
mTOR-inhibitors	1.9	0.5–7.2	0.319	2.8	0.5–14.5	0.225
Corticosteroids	0.7	0.3–1.6	0.374	0.6	0.2–1.6	0.289
Rejection treated with pulse-steroid	0.7	0.3–2.0	0.550	0.8	0.3–2.2	0.622

* In adjusted models one variable was tested at a time while adjusting for age, sex, and transplant type. Ref., Reference; HBV, hepatitis B virus.

**Table 4 vaccines-12-01092-t004:** Sensitivity analysis. Risk of non-response to vaccination among SOT recipients with serology at time of listing for transplantation and post-transplantation serology.

	Unadjusted Model			Adjusted Model *		
	OR (Crude)	95% CI	*p*-Value	aOR (Adjusted)	95% CI	*p*-Value
**HBV (*n* = 122)**						
Age at time of listing for transplantation per 10-year increase	1.5	1.1–2.1	0.009			
Male sex	0.8	0.3–1.7	0.496			
Transplant type						
Liver	Ref.	Ref.	Ref.			
Heart	0.6	0.2–1.9	0.349			
Lung	1.3	0.4–4.0	0.614			
Anti-HBs at time of listing for transplantation						
Unprotected	Ref.	Ref.	Ref.	Ref.	Ref.	Ref.
Protected	0.1	0.0–0.1	<0.001	0.04	0.0–0.1	<0.001
**Invasive pneumococcal disease (*n* = 132)**						
Age at time of listing for transplantation per 10-year increase	1.1	0.8–1.5	0.494			
Male sex	0.4	0.2–0.9	0.020			
Transplant type						
Liver	Ref.	Ref.	Ref.			
Heart	1.2	3.8	0.707			
Lung	1.3	0.6–2.9	0.567			
Pneumococcal antibodies at time of listing for transplantation						
Unprotected	Ref.	Ref.	Ref.	Ref.	Ref.	Ref.
Protected	0.4	0.2–0.9	0.017	0.3	0.1–0.7	0.007

* In adjusted models one variable was tested at a time while adjusting for age, sex, and transplant type. Ref., Reference; SOT, solid organ transplant; HBV, hepatitis B virus; anti-HBs, hepatitis B surface antibodies.

**Table 5 vaccines-12-01092-t005:** Serology for pneumococcal antibodies.

	Protected (*n* = 58)	Unprotected (*n* = 78)	*p*-Value *
Age at serology (median (IQR) in years)	55 (46.2, 61.8)	55.5 (49.2, 61.0)	0.827
Age at time of listing for transplantation (median (IQR) in years)	53.5 (44.2, 60.0)	53 (48, 59)	0.708
Male sex, *n* (%)	40 (69.0)	39 (50.0)	0.041
Transplant type, *n* (%)			0.786
Liver	39 (67.2)	48 (61.5)	
Lung	13 (22.4)	21 (26.9)	
Heart	6 (10.3)	9 (11.5)	
Re-transplantation, *n* (%)	<5	5 (8.6)	NA
Pneumococcal antibodies at time of listing for transplantation, *n* (%)			0.026
Unprotected	32 (56.1)	57 (76.0)	
Protected	25 (43.9)	18 (24.0)	
Missing	1	3	
Pneumococcal vaccination status, *n* (%)			0.008
Completed pre-transplantation	44 (75.9)	52 (66.7)	
Completed after transplantation	12 (20.7)	26 (33.3)	
Incomplete	<5	<5	
Time from transplantation to serology (median (IQR) in days)	696 (444.8, 977.8)	554.5 (298.2, 870.2)	0.103
Time from vaccination to serology (median (IQR) in days)	792 (584.2, 1104.5)	701 (497.2, 1043.5)	0.122
Time from vaccination to transplantation (median (IQR) in days)	116 (57.2, 194.5)	120 (54.0, 233.8)	0.639
Comorbidities after transplantation, *n* (%)			
Diabetes mellitus	15 (25.9)	14 (17.9)	0.367
Dialysis	<5	<5	NA
Cancer	<5	<5	NA
Immunosuppressive therapy			
Immunosupp. at time of listing for transplantation, *n* (%)	21 (36.2)	23 (29.5)	0.520
Induction therapy, *n* (%)			0.874
No induction	34 (58.6)	43 (55.1)	
Simulect/basiliximab	12 (20.7)	16 (20.5)	
ATG	12 (20.7)	19 (24.4)	
Maintenance immunosuppresive therapy, *n* (%)			
Corticosteroid	36 (62.1)	55 (70.5)	0.395
Tacrolimus	45 (77.6)	59 (75.6)	0.952
Cyclosporine	9 (15.5)	17 (21.8)	0.484
mTOR-inhibitors	8 (13.8)	8 (10.3)	0.716
Antimetabolites			0.953
Azathioprine	<5	5 (6.4)	
Mycophenolate mofetil	47 (81.0)	64 (82.1)	
None	7 (12.1)	9 (11.5)	
Rejection treated with pulse-steroid, *n* (%)	6 (10.3)	15 (19.2)	0.239

IQR, interquartile range; NA, not applicable; ATG, antithymocyte globulin. * Calculated by chi-square test or Mann–Whitney U test.

**Table 6 vaccines-12-01092-t006:** Risk factors for non-response to pneumococcal vaccination.

	Unadjusted Model			Adjusted Model *		
	OR (Crude)	95% CI	*p*-Value	OR (Adjusted)	95% CI	*p*-Value
Age at the time of listing for transplantation per 10-year increase	1.1	0.8–1.5	0.530			
Male sex	0.5	0.2–0.9	0.028			
Transplant type						
Liver	Ref.	Ref.	Ref.			
Heart	1.2	0.4–3.7	0.728			
Lung	1.3	0.6–3.0	0.511			
Time variables						
Time from transplantation to serology per year	0.7	0.5–1.1	0.115	0.7	0.5–1.1	0.116
Time from vaccination to transplantation per 3 months	1.1	0.9–1.3	0.519	1.1	0.8–1.4	0.546
IPD vaccination status						
Completed pre-transplantation	Ref.	Ref.	Ref.	Ref.	Ref.	Ref.
Completed after transplantation	1.8	0.8–4.1	0.134	1.9	0.8–4.4	0.153
Incomplete	-	-	-	-	-	-
Comorbidities after transplantation						
Diabetes mellitus	0.6	0.3–1.4	0.267	0.7	0.3–1.9	0.515
Immunosuppress. therapy after transplantation						
Antimetabolites						
None	Ref.	Ref.	Ref.	Ref.	Ref.	Ref.
Azathioprine	1.0	0.2–5.0	0.973	1.0	0.2–5.4	0.963
Mycophenolate mofetil	1.1	04–3.1	0.915	1.3	0.4–3.9	0.646
Calcineurin inhibitors						
Tacrolimus	0.9	0.4–2.0	0.791	1.0	0.4–2.8	0.957
Cyclosporine	1.5	0.6–3.7	0.359	1.5	0.5–4.8	0.533
mTOR-inhibitors	0.7	0.3–2.0	0.528	0.7	0.2–2.0	0.513
Corticosteroids	1.5	0.7–3.0	0.302	1.2	0.5–2.6	0.725
Rejection treated with pulse-steroid	2.1	0.7–5.7	0.162	2.2	0.8–6.3	0.149

Ref., Reference; * In adjusted models one variable was tested at a time while adjusting for age, sex, and transplant type. IPD, invasive pneumococcal disease.

## Data Availability

The data that support the findings of this study are available from the corresponding author upon reasonable request.

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
