# Peer review of "Post-Transplantation Seroprotection Rates in Liver, Lung, and Heart Transplant Recipients Vaccinated Pre-Transplantation against Hepatitis B Virus and Invasive Pneumococcal Disease"

_vaccines, 2024, doi:10.3390/vaccines12101092_

Round 1
Reviewer 1 Report
Comments and Suggestions for Authors
Dr. Lise Bank Hornung et al. have reported that the importance of timing the administration of hepatitis B virus (HBV) and pneumococcal vaccines to solid organ transplant (SOT) recipients.
I have some questions and comments, however, that I believe the authors need to address, as follows:
<Major criticisms>
1. I think this needs to be corrected, as one table is spread over several pages and is difficult to read (Table1-5).
2. If possible, a discussion of the mechanism by which the timing of vaccines administration changes the likelihood of antibody production should be conducted using previous papers.
<Minor criticisms>
1. The explanation of the HBV abbreviation should be placed on line 28, not line 32.
2. The authors should mention about the abbreviation of Page 13, line 257: ‘HIV’.
Author Response
Dr. Lise Bank Hornung et al. have reported that the importance of timing the administration of hepatitis B virus (HBV) and pneumococcal vaccines to solid organ transplant (SOT) recipients.
Thank you for taking you time to review our manuscript. We appreciate your comments and hope that you find the revised manuscript suitable for publication.
I have some questions and comments, however, that I believe the authors need to address, as follows:
Major criticisms:
- I think this needs to be corrected, as one table is spread over several pages and is difficult to read (Table1-5).
Thank you very much for pointing this out. We agree with this comment. We have made the layout of the tables more clear. When one table spreads over more than one page we have added a second legend, to inform the reader that the table is a continuation of the table on the previous page. - If possible, a discussion of the mechanism by which the timing of vaccines administration changes the likelihood of antibody production should be conducted using previous papers.
Thank you very much for this interesting comment. We agree with this suggestion and we have added this to the discussion.
Changes to the manuscript
We have added the following to the discussion section in line 240:
“Furthermore, we found an association between longer time from assessment at the vaccination clinic to transplantation and lower risk of HBV non-response. This could be due to increased likelihood of completing the vaccination schedule before transplantation. Another explanation could be an association between longer time from assessment at the vaccination clinic to transplantation and less severe organ disease, which has been described to be associated with lower risk of non-response in liver transplant recipients​19.”
The following has been added to the results section, line 178
“Furthermore, SOT recipients with longer time from assessment at the vaccination clinic to transplantation had an OR of 0.7 ([95% CI: 0.6-0.9], p=0.008) for non-response per 3 months of increase in time from assessment at the vaccination clinic to transplantation.”
The following has been added to the results section, line 182
“SOT recipients with longer time from assessment at the vaccination clinic to transplantation had an aOR of 0.8 ([95% CI: 0.6-1.0], p=0.043) for non-response per 3 months of increase in time from assessment to transplantation (table 3).”
Minor criticisms
- The explanation of the HBV abbreviation should be placed on line 28, not line 32.
Thank you very much for pointing this out. We have revised as suggested and explained the abbreviations "HBV" and "IPD" in the abstract in line 28, which previously read:
“We aimed to investigate the proportion of transplant recipients vaccinated against HBV and IPD pre-transplantation at the time of listing for transplantation with seroprotection post-transplantation.”
It now reads:
“We aimed to investigate the proportion of transplant recipients vaccinated against hepatitis B virus (HBV) and invasive pneumococcal disease (IPD) pre-transplantation at the time of listing for transplantation with seroprotection post-transplantation.” - The authors should mention about the abbreviation of Page 13, line 257: ‘HIV’.
Thank you very much for pointing this out. We have revised as suggested and explained the abbreviation "HIV" in line 257, which previously read:
“A suggestion for further investigations is double dose HBV vaccination, which, in a randomized controlled trial, had an OR for seroprotection of 2.48 compared to single dose when administered to 107 non-responding patients with HIV.[27]”
It now reads:
”A suggestion for further investigations is double dose HBV vaccination, which, in a randomized controlled trial, had an OR for seroprotection of 2.48 compared to single dose when administered to 107 non-responding patients with human immunodeficiency virus (HIV)[27].”
Reviewer 2 Report
Comments and Suggestions for Authors
Authors investigated post-transplantation antibody concentrations against hepatitis B virus (HBV) and invasive pneumococcal disease (IPD). Based on the comparison before and after SOT, they argue that early vaccination at the time of listing is important for high seroprotection against HBV and IPD. In addition, not only initiating, but also completing the schedule prior to transplantation mattered. It was suggested that pre-transplantation seroprotection may be an important factor when aiming to achieve seroprotection after transplantation. Also, the risk factors for responses below the protective threshold (non-response) was determined.
Despite the important nature of these findings, the following issues must be addressed before publication.
The meaning of Figure 2 must be explained and the need for further analysis.
Additional figures or diagrams summarizing the findings would improve the manuscript.
Minor Points:
1. Line 29: IPD must be explained where it first appears (invasive pneumococcal disease).
2. LIne 164: patient -> Patient
3. In Tables 3, 5, and 6, the adjusted model must be explained in more detail, including how it was adjusted for age, sex, and transplant type.
Comments on the Quality of English LanguageFor better reading, the manuscript must be edited by a native speaker of English.
Author Response
Authors investigated post-transplantation antibody concentrations against hepatitis B virus (HBV) and invasive pneumococcal disease (IPD). Based on the comparison before and after SOT, they argue that early vaccination at the time of listing is important for high seroprotection against HBV and IPD. In addition, not only initiating, but also completing the schedule prior to transplantation mattered. It was suggested that pre-transplantation seroprotection may be an important factor when aiming to achieve seroprotection after transplantation. Also, the risk factors for responses below the protective threshold (non-response) was determined.
Despite the important nature of these findings, the following issues must be addressed before publication.
Thank you for taking you time to review our manuscript. We appreciate your comments and hope that you find the revised manuscript suitable for publication.
Major points:
- The meaning of Figure 2 must be explained and the need for further analysis
Thank you very much for this comment. We agree and have added further explanation to the legend for figure 2, which previously read:
“Figure 2. Seroprotection rates against hepatitis B virus (HBV) and invasive pneumococcal disease (IPD) at time of listing for transplantation and post-transplantation. SOT, solid organ transplant.”
It now reads:
“Figure 2. Seroprotection rates among SOT candidates and recipients vaccinated against hepatitis B virus (HBV) and invasive pneumococcal disease (IPD) at time of listing for transplantation. Dark orange: Percentage of SOT candidates and recipients with seroprotection. Light orange: Percentage of SOT candidates and recipients without seroprotection. X-axis: Seroprotection rates at the time of listing for transplantation (two columns on the left) and seroprotection rates post-transplantation (two columns on the right). Y-axis: Percentage of SOT candidates and recipients. SOT, solid organ transplant.”
- Additional figures or diagrams summarizing the findings would improve the manuscript.
Thank you very much for this suggestion. We agree with the comment. Therefore, we have included a textbox in line 294 with a bullet-point summary of the most important findings of the text:
”Summary of key findings
- Post-transplantation seroprotection rates for both hepatitis B virus (HBV) and invasive pneumococcal disease (IPD) were 9% (n=38) and 42.6% (n=58) , respectively.
- Risk factors for HBV non-response were completed HBV vaccination post-transplantation, incomplete HBV vaccination, and shorter time from assessment at the vaccination clinic to transplantation.
- SOT recipients with seroprotection at the time of listing for organ transplantation had lower odds of non-response, suggesting early vaccination should be a priority.”
Minor Points:
- Line 29: IPD must be explained where it first appears (invasive pneumococcal disease).
Thank you very much for pointing this out. We have revised as suggested and explained the abbreviations "HBV" and "IPD" in the abstract in line 28 and 29, which previously read:
“We aimed to investigate the proportion of transplant recipients vaccinated against HBV and IPD pre-transplantation at the time of listing for transplantation with seroprotection post-transplantation.”
It now reads:
“We aimed to investigate the proportion of transplant recipients vaccinated against hepatitis B virus (HBV) and invasive pneumococcal disease (IPD) pre-transplantation at the time of listing for transplantation with seroprotection post-transplantation.”
- LIne 164: patient -> Patient
Thank you very much for pointing this out. We agree and have made the suggested revision in line 164, which previously read:
“Table 1. patient characteristics at time of listing for transplantation.”
And now reads:
“Table 1. Patient characteristics at time of listing for transplantation.”
- In Tables 3, 5, and 6, the adjusted model must be explained in more detail, including how it was adjusted for age, sex, and transplant type.
Thank you for pointing out the need for further explanation of the adjusted models in the tables. We have made changes accordingly as follows:
The table footnote that previously read:
“*adjusted for age, sex, and transplant type.”
Now reads:
“* In adjusted models one variable was tested at a time while adjusting for age, sex, and transplant type.”
Reviewer 3 Report
Comments and Suggestions for Authors
The manuscript entitled: “Post-transplantation seroprotection rates in liver, lung, and heart transplant recipients vaccinated pre-transplantation against hepatitis B virus and invasive pneumococcal disease" is well designed and written. Nowadays, when we have more and more transplants and subsequent immunosuppression, the importance of vaccination in these patients is even more pronounced. The goal of the work is very important to assess how many patients are protected from possible severe infections caused by HBV and invasive pneumococcal diseases. The manuscript composes all parts of the scientific work. The methodology is clearly written with the use of adequate scientific terms. The results are presented in a clear way, only the tables shown in results section are little to big and difficult to follow. Also in the results in the textual part, Table 6 appears after Table 3, this should be corrected and the tables should be marked in the order in which they appear in the paper. The discussion is clearly written and follows the results. You also listed the limitations of the study emphasis that further research on this topic are possible and needed in order to have a more complete picture of a very complex topic such as the success of vaccination in transplanted patients.
Some small mistakes in writing are present in the text and names of the tables and figures that need to be corrected.
Comments on the Quality of English LanguageSmall typing errors nothing bigger than that.
Author Response
The manuscript entitled: “Post-transplantation seroprotection rates in liver, lung, and heart transplant recipients vaccinated pre-transplantation against hepatitis B virus and invasive pneumococcal disease" is well designed and written. Nowadays, when we have more and more transplants and subsequent immunosuppression, the importance of vaccination in these patients is even more pronounced. The goal of the work is very important to assess how many patients are protected from possible severe infections caused by HBV and invasive pneumococcal diseases. The manuscript composes all parts of the scientific work. The methodology is clearly written with the use of adequate scientific terms. The results are presented in a clear way, only the tables shown in results section are little to big and difficult to follow. Also in the results in the textual part, Table 6 appears after Table 3, this should be corrected and the tables should be marked in the order in which they appear in the paper. The discussion is clearly written and follows the results. You also listed the limitations of the study emphasis that further research on this topic are possible and needed in order to have a more complete picture of a very complex topic such as the success of vaccination in transplanted patients.
Some small mistakes in writing are present in the text and names of the tables and figures that need to be corrected.
Thank you for taking you time to review our manuscript. We appreciate your comments and hope that you find the revised manuscript suitable for publication.
- The results are presented in a clear way, only the tables shown in results section are little to big and difficult to follow. Also in the results in the textual part, Table 6 appears after Table 3, this should be corrected and the tables should be marked in the order in which they appear in the paper.
Some small mistakes in writing are present in the text and names of the tables and figures that need to be corrected.
Thank you very much for pointing this out. We agree with this comment. We have made the layout of the tables more clear, and numbered and moved the tables to appear correctly after the data is mentioned in the results section. When a table spreads over more than one page we have added a second legend, to inform the reader that the table is a continuation of the table on previous page.